# Postmortem high-dimensional immune profiling of severe COVID-19 patients reveals distinct patterns of immunosuppression and immunoactivation

Haibo Wu[1,11], Peiqi He[2,3,4,11], Yong Ren[5,11], Shiqi Xiao[5], Wei Wang[1], Zhenbang Liu[2], Heng Li[1], Zhe Wang[1], Dingyu Zhang[6], Jun Cai[7], Xiangdong Zhou[8], Dongpo Jiang[8], Xiaochun Fei[7], Lei Zhao[7], Heng Zhang[7], Zhenhua Liu[7], Rong Chen[6], Weiqing Li[9], Chaofu Wang[7], Shuyang Zhang[10], Jiwei Qin[4], Björn Nashan [ID] [4] & Cheng Sun [ID] [2,3,4✉]

A complete diagnostic autopsy is the gold-standard to gain insight into Coronavirus disease 2019 (COVID-19) pathogenesis. To delineate the in situ immune responses to SARS-CoV-2 viral infection, here we perform comprehensive high-dimensional transcriptional and spatial immune profiling in 22 COVID-19 decedents from Wuhan, China. We find TIM-3-mediated and PD-1-mediated immunosuppression as a hallmark of severe COVID-19, particularly in men, with PD-1[+] cells being proximal rather than distal to TIM-3[+] cells. Concurrently, lymphocytes are distal, while activated myeloid cells are proximal, to SARS-CoV-2 viral antigens, consistent with prevalent SARS-CoV-2 infection of myeloid cells in multiple organs. Finally, viral load positively correlates with specific immunosuppression and dendritic cell markers. In summary, our data show that SARS-CoV-2 viral infection induces lymphocyte suppression yet myeloid activation in severe COVID-19, so these two cell types likely have distinct functions in severe COVID-19 disease progression, and should be targeted differently for therapy.

[1] Department of Pathology, the First Affiliated Hospital of USTC, Division of Life Sciences and Medicine, University of Science and Technology of China, Hefei, Anhui 230036, China. [2] CAS Key Laboratory of Innate Immunity and Chronic Disease, School of Basic Medical Sciences, Division of Life Sciences and Medicine, University of Science and Technology of China, Hefei 230027, China. [3] Institute of Immunology, University of Science and Technology of China, Hefei 230027, China. [4] Transplant & Immunology Laboratory, the First Affiliated Hospital of USTC, Division of Life Sciences and Medicine, University of Science and Technology of China, Hefei, Anhui 230001, China. [5] Department of Pathology, the First Hospital Affiliated to Army Medical University, Chongqing 400038, China. [6] Wuhan Jinyintan Hospital, Wuhan, Hubei 430015, China. [7] Shanghai Jiaotong University School of Medicine, Shanghai 200030, China. [8] Third Military Medical University Daping Hospital, Chongqing 400038, China. [9] Department of Critical Care Medicine, Key Laboratory of Emergency and Critical Care Research, Jinling Hospital, Nanjing University, Nanjing, Jiangsu 210002, China. [10] Peking Union Medical College Hospital, Peking 100730, China. [11]These authors contributed equally: Haibo Wu, Peiqi He, Yong Ren. ✉email: charless@ustc.edu.cn

Since the COVID-19 outbreak in January 2020, researchers have identified that immune dysregulation with lymphopenia and cytokine storm are hallmarks of severe disease[1]. However, much remains unknown about how this virus interacts with the human immune system. Most attempts to profile immune responses have been performed with peripheral blood samples[2], but direct pathological evidence is critical to delineate the underlying pathophysiological mechanisms. Pathological reports are still limited by small sample sizes owing to onerous safety requirements and insufficient political support for performing autopsies[3]. Moreover, many prior studies have only utilized low throughput techniques like traditional immunohistochemistry (IHC) and PCR (Supplementary Table 1): these methods are inadequate for obtaining a complete picture of the heterogeneous immune landscape.

We performed a comprehensive, high-dimensional analysis of the immune signature in 22 decedents (53–88 years old) who had contracted the disease in Wuhan, China and succumbed to COVID-19 after experiencing a severe infection (Supplementary Table 2). We performed transcriptome analyses, Digital Spatial Profiling (DSP), and multiplex-IHC (mIHC) of autopsy tissue samples collected from patients admitted to the Huoshenshan Hospital of Wuhan, China in March 2020.

Here, using a combination of transcriptome analyses with DSP and mIHC, we report the high-dimensional immunopathological profiling of severe COVID-19 patients in multiple organs, including various immune activation-related markers (CD69, IFN-γ, granzyme B, etc.) and immune suppression and exhaustion related markers (TIM-3, PD-1, BTLA, NKG2A, etc.). We highlight the distinct patterns of severe immunosuppression in multiple organs and find TIM-3 and PD-1-mediated immunosuppression is a hallmark of severe COVID-19, and that the extent of immunosuppression correlates with male sex and advanced age. Particularly, activated myeloid cells, but not lymphocytes, are proximity to SARS-CoV-2 viral antigen, consistent with prevalent SARS-CoV-2 infection of myeloid cells in multiple organs. Our data show the COVID-19 infection landscape across multiple organs and demonstrate the viral load is positively associated with ACE2 expression as well as expression of markers of immunosuppression. In conclusion, we suggest that SARS-CoV-2 viral infection induces lymphocyte suppression yet myeloid activation in severe COVID-19. We should thus consider the two immune compartments separately when investigating disease progression.

## Results

**Severe COVID-19 infection is characterized by immunosuppression.** To examine the associations between COVID-19 disease and patient characteristics, we performed bulk RNA-sequencing on lung samples from 11 COVID-19 decedents and 3 non-COVID-19 decedents. We detected clear differences in the transcriptomic profiles between COVID-19 decedents and non-COVID-19 decedents (Fig. 1a–b, Supplementary Fig. 1a–b). By gene set enrichment analysis (GSEA), we determined that the corresponding enriched pathways involved negative regulation of the immune response and reduced leukocyte activation in COVID-19 decedents (Fig. 1c). Interestingly, transcriptomic profiles of the lung tissues were more similar within than between the sexes (Fig. 1d, Supplementary Fig. 1c), suggesting that sex might influence the immune landscape in COVID-19 infection. This trend of sex differences has been further observed in the liver, kidney, spleen, and heart tissues (Fig. 1d). In particular, the immunosuppression signature was more pronounced in men compared to women (Fig. 1e–f) across all organs under consideration and in this subset of decedents translated to a shorter survival duration in men (Supplementary Fig. 1d).

**TIM-3 and PD-1 mediate immunosuppression in men.** To identify the sex-biased molecular mediators of immunosuppression in COVID-19 infection, we transcriptionally profiled the in situ immune environments of each sample (Fig. 2a). Corroborating the findings from bulk RNA-sequencing, we found that lung tissue from men expressed higher levels of inhibitory receptors, including *HAVCR2* (TIM-3), *PDCD1* (PD-1), *ENTPD1* (CD39), *BTLA* and KIRs compared to women, but lower levels of immune activation-related markers, such as *MKI67* (Ki-67), and *CD69* (Fig. 2b–c). These trends were partially recapitulated in the heart and kidney but different from what was seen in the liver (Supplementary Fig. 2a–b). To validate the immunosuppressive signature at the protein level, we performed mIHC and similarly found that higher levels of TIM-3$^+$ and PD-1$^+$TIM-3$^+$ cells were observed in the lung and kidney tissue of men compared to women but that the reverse was relationship observed in the liver (Fig. 2d, Supplementary Fig. 2c–d). In addition, we observed a positive association between lung TIM-3$^+$ cells and age in men but not in women (Supplementary Fig. 2e).

Because clustering of multiple suppressive markers can compound immunosuppression[4], we scrutinized the microenvironment surrounding these TIM-3$^+$ cells by investigating their spatial relationship with other immunosuppressive markers. Interestingly, we observed that in the lung, liver, and kidney, the PD-1$^+$ cells were significantly more likely to be proximal than distal from TIM-3$^+$ cells (Fig. 2e–f). These findings suggest an association between TIM-3 and PD-1 upregulation and an unfavorable clinical outcome, and the potential use of anti-TIM-3 and/or anti-PD-1 therapy in combating COVID-19.

**Lymphocytes are selectively suppressed in COVID-19.** To identify the immune compartment affected by TIM-3 and PD-1-mediated immunosuppression, we separately compared CD3$^+$- and CD68$^+$ -rich regions (Fig. 3a, Supplementary Fig. 3a). In CD3$^+$ -rich regions of the lung, we found a significantly higher expression of immunosuppression-associated genes and various immunosuppressive cytokines in men than in women (Fig. 3b–c). We observed a similar trend in the CD68$^+$-rich region, although it did not statistically differ between the sexes (Supplementary Fig. 3b). We also found higher expression of genes associated with innate immune cells (*CD83*, *NKG7* and *CD68*) and natural killer (NK) cell activation (*GZMB, KLRK1, KLRF1, KLRC2* and *NCR1*) in the lung CD3$^+$-rich regions of men compared to women (Supplementary Fig. 4a). Thereafter, to verify the gender biased immunosuppression in other organs, we compared the expression of genes associated with immune cells, immune activation, cytokine (Supplementary Fig. 4b–e) and immunosuppression (Supplementary Fig. 5a) from CD3$^+$- and CD68$^+$ -rich regions in the liver, kidney, heart and spleen. We observed similar trends in the CD3$^+$- and CD68$^+$-rich region in the heart, spleen and kidney, but not in the liver (Supplementary Fig. 5a). The difference in immunosuppression-associated gene expression such as *PDCD1* in CD3$^+$-rich regions was also more significant than the difference in CD68$^+$-rich region in the heart, spleen and kidney, but not in the liver (Supplementary Fig. 5b–e).

We next asked if the immunosuppressive profile of COVID-19 decedents would manifest in the microenvironment surrounding SARS-CoV-2 viral antigen$^+$ cells. To this end, we characterized the spatial relationship between SARS-CoV-2 nucleocapsid protein (NP) and surrounding cell types. We observed that activated GZMB$^+$CD3$^+$ cells did not exhibit any association with proximity to viral NP in the lung, kidney, heart, and spleen but were associated with proximity to viral NP in the liver (Fig. 3d–e). In lung and spleen tissue, NK cells and CD8$^+$ cells were less likely to be proximal to viral NP whereas the reverse relationship was observed for the liver and heart (Fig. 3d–e). Together, these

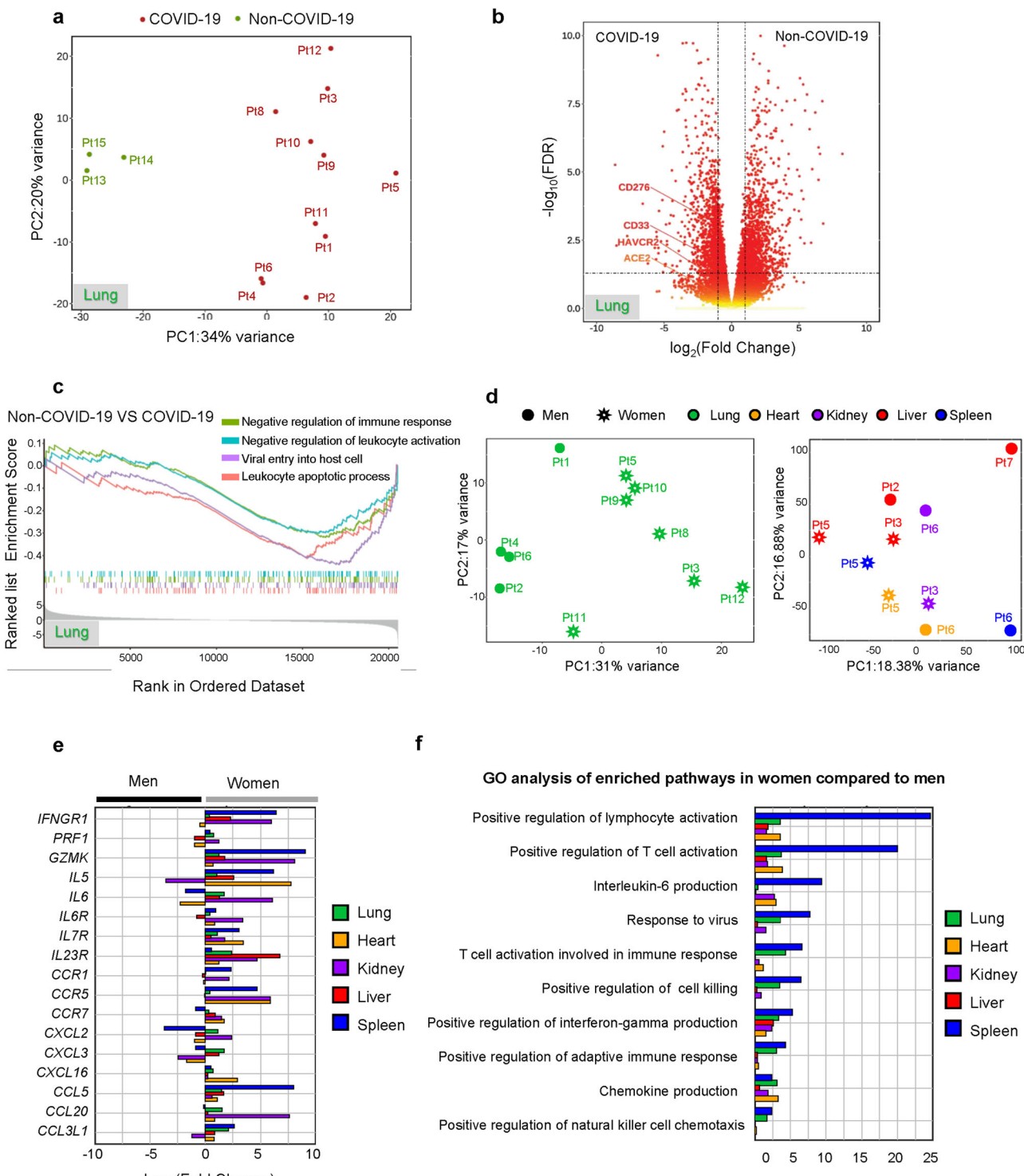

**Fig. 1 Severe COVID-19 infection is characterized by immunosuppression and is more pronounced in men than women. a** Principal component analysis of lung tissue of COVID-19 ($n = 11$) compared to non-COVID-19 ($n = 3$) decedents. Each dot corresponds to a single sample based on RNA-seq data. **b** The differential genes expression between COVID-19 and non-COVID-19 of lung tissue, the vertical baseline represents the absolute value of $\log_2$ fold change >1, and the horizontal baseline above represents $p < 0.05$. DESeq2 was used for differential gene expression analysis. **c** Gene set enrichment analysis of hallmark lymphocyte inhibitory genes based on non-COVID-19 to COVID-19. FDR < 0.05 defined as significant. Gene sets are ranked by the difference in the fraction of experiments with significant positive and negative enrichment. **d** Principal component analysis of bulk transcriptomes of lung (green, women = 7, men = 4), heart (yellow, women = 1, men = 1), kidney (purple, women = 1, men = 1), liver (red, women = 2, men = 2) and spleen (blue, women = 1, men = 1). **e** Comparison chemokine expression levels between men and women in lung, heart, kidney, liver and spleen. Colored according to the sample based on RNA-seq data. **f** Gene Ontology analysis of pathways enriched of multiple organs in women upregulate compared to men. Colored according to the sample based on RNA-seq data. Fisher's exact test. Source data are provided as a Source Data file.

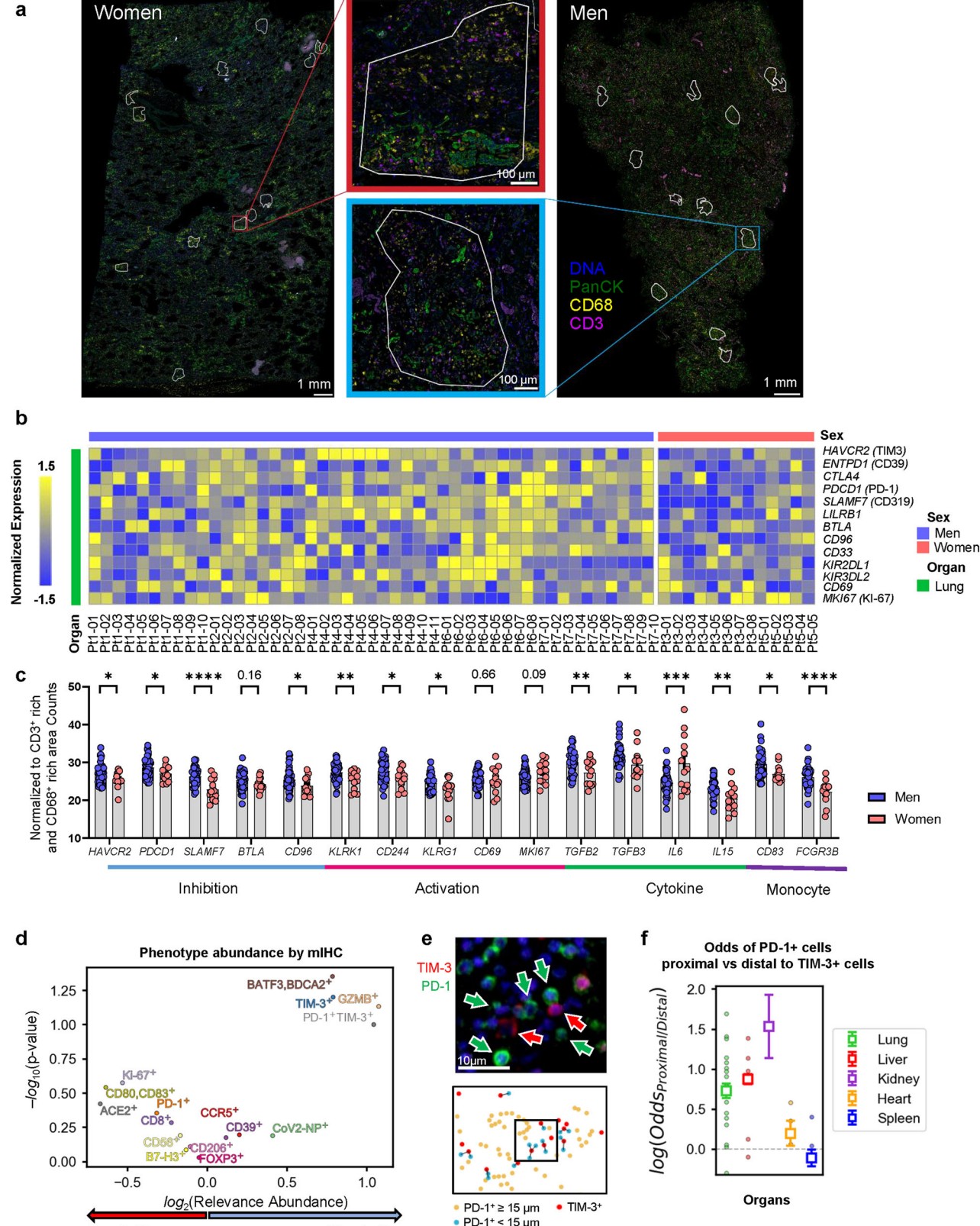

suggest that infected cells can escape detection by GZMB⁺CD3⁺ cells, NK cells, and CD8⁺ T-cells, particularly in the lung.

Notably, the myeloid population behaved differently as activated myeloid cells [CD80/83⁺ CD206⁺ macrophages and BDCA2⁺/BATF3⁺ dendritic cells (DCs)] were significantly associated with proximity to viral NP⁺ cells in the lung, liver,

heart, and kidneys (Fig. 3f–g). Furthermore, the myeloid cells proximal to viral NP⁺ cells were more likely to be activated than those distal from viral NP⁺ cells across most organs (Fig. 3h). These findings indicate that severe COVID-19 might selectively suppress lymphocytes, particularly in the lung, while myeloid cells retain their capacity to mount an immune response.

**Fig. 2 Men exhibit a greater extent of COVID-19-associated immunosuppression than women. a** Representative tissue sections from men (right) and women (left) of lung with 12 selected ROIs per section (white outlines) analyzed by DSP. **b** Heatmap show normalized gene expression levels across 60 ROIs from7 COVID-19 decedents (men = 2, women = 5) of lung. Z -scale from −1.5 to 1.5. **c** Comparison of expression of selected genes between sexes of lung tissues with COVID-19. Each point represents one ROI. Colored dots represent different samples. Blue (n = 47), red (n = 13). Data are shown as mean ± s.e.m. p (HAVCR2) = 0.04; p (PDCD1) = 0.03; p (SLAMF7) = 0.00003; p (CD96) = 0.049; p (KLRK1) = 0.006; p (CD244) = 0.02; p (KLRG1) = 0.03; p (TGFB2) = 0.006; p (TGFB3) = 0.04; p (IL6) = 0.0003; p (IL15) = 0.002; p (CD83) = 0.01; p (FCGR3B) = 0.00002; Unpaired two-tailed Student's t tests. *p < 0.05, **p < 0.01, ***p < 0.001, ****p < 0.0001, ns: not significant. **d** Comparison of percentages of cells in lung tissue expressing various immune markers between men (n = 11) and women (n = 10). CoV2-NP: SARS-CoV-2 nucleocapsid protein. p value: two-tailed Student's t test. **e** mIHC staining (top) and annotation (bottom) of PD-1+ (green arrows) cells proximal to TIM-3+ cells (red arrows) in lung tissue. Depicted here are 5 out of the 1357 PD-1+ cells and 2 out of the 13682 TIM-3+ cells observed in lung tissue sections. **f** Odds ratio of TIM-3+ cell-proximal compared to TIM-3+ cell-distal PD-1+ cells in various organs. Odds ratio defined as $(N_{(p)PD1+}/N_{(d)PD1+})/(N_{(p)PD1−}/N_{(d)PD1−})$, where (p) and (d) refer to cells proximal or distal from TIM-3+ cells. Proximity is as defined by a distance cut-off of 15 μm. Data are shown as aggregate log(Odds Ratio) ± 95% CI.(Lung: 0.724 ± 0.092; Liver: 0.873 ± 0.062; Kidney: 1.532 ± 0.397; Heart: 0.195 ± 0.154; Spleen: 0.111 ± 0.104). Dots indicate sample odds ratios, squares with error bars indicate aggregated odds ratios and confidence intervals. Source data are provided as a Source Data file.

**ACE2 expression is associated to higher infection rate**. Having established selective lymphocyte suppression in situ during severe COVID-19, we sought to identify additional correlates of viral load by stratifying the regions of interest based on the expression levels of SARS-CoV-2 ORF1ab. In doing so, we validated that PDCD1, BTLA, TIGIT and CD96 were highly expressed in virus-high regions, especially in the lungs, livers and spleens (Fig. 4a–b, Supplementary Fig. 6), lending support to our hypothesis that viral presence drives selective lymphocyte immunosuppression in situ. Reflecting its role in viral infection, ACE2 expression was higher in virus-high regions compared to virus-low regions of the lung, liver and spleen (Fig. 4b). This trend of expression is not significantly different in the heart and kidneys. We next characterized the distribution of viral antigens amongst various cell types and determined that in the lung, liver, and kidney and spleen, the predominant NP+ cells were DCs (BATF3/BDCA2+) and furthermore that macrophages (CD206+) constituted an appreciable fraction of NP+ cells only in the liver (Fig. 4c). We also discovered that BATF3/BDCA2+ DCs exhibited a higher tendency for infection over BATF3/BDCA2− cells in all organs under consideration (Fig. 4e) and that only in the liver, CD206+ macrophages were more likely to be infected than CD206− cells while in the lungs and heart, the converse was true (Fig. 4d). In the liver, BATF3/BDCA2+ DCs that were ACE2+ exhibited an even higher likelihood of infection than ACE2− BATF3/BDCA2+ DCs (Fig. 4d–e).

In sum, we characterized the COVID-19 infection landscape across multiple organs and found ACE2 expression to be correlated with infection susceptibility, and viral load to be positively associated with ACE2 expression as well as expression of markers of immunosuppression.

## Discussion

COVID-19 infection elicits robust T-cell responses;[5] however, poor clinical outcomes have been associated with lymphopenia[1] and an increased number of non-functional T cells[6]. These findings expose the central role of the adaptive immune response in viral clearance. In this study, we suggest that beyond lymphopenia, selective lymphocyte immunosuppression drives disease pathology, particularly in men with severe COVID-19.

The higher morbidity and mortality rates described in men[1, 2, 7, 8] follows a common theme in viral infections. Due to biological differences involving estrogen regulation of X-linked genes, incomplete X-inactivation, and modulation of interferon production by sex steroids[9, 10], men may experience a higher case fatality rate than women. Based on the results of our study, we propose male-biased immunosuppression as an additional possible mechanism that accelerates disease severity. Specifically, we found in the tissue context that this mechanism operates via

immune checkpoint markers on T cells and NK cells but not on DCs and macrophages.

Previous reports have described upregulated CD39, PD-1, BTLA or TIM-3 expression in T cells and NK cells in the peripheral blood of severe COVID-19 patients[4, 11–16]. Complementing these reports within the tissue context, we found in the organs autopsied that viral load correlated with immunosuppressive marker expression and that the immunosuppressive signature is more pronounced in men than women. We additionally identified a greater clustering of PD-1+ cells around TIM-3+ cells in men than women, which compounds the degree of inhibition of lymphocytes, as multiple marker expression parallels suppression severity[4]. Furthermore, severe COVID-19 has also been associated with non-functional T cells[6] and a less expanded T-cell population[17]. As we detected a higher abundance of lymphocytes distant from rather than close to infected cells, the functionality of T cells is likely to be reined in by local virus-induced expression of immunosuppressive markers.

Taken together, these observations suggest consideration of the use of immunotherapy strategies such as co-blockade of PD-1 and TIM-3, especially in men, to reinvigorate the immune system against COVID-19 infection. One argument against the use of such checkpoint inhibitors is that it might upset the balance between viral clearance and immunopathology caused by the excessive T-cell response[18]. Several case reports involving cancer patients noticed severe COVID-19 development alongside treatment with checkpoint inhibitors[18–21]. However, with cancer as a co-morbidity, these patients are more susceptible to disease complications[22] and therefore these reports remain inconclusive until larger studies are conducted[17]. Mounting evidence also suggests that the timing of administration of therapeutics might be a key determinant of immunopathological response[23].

A common complication of severe COVID-19 is that of cytokine release syndrome, in which plasma levels of cytokines such as IL-2, IL-6, IFN-γ, and TNF-α are elevated[24, 25]. In the context of COVID-19, this phenomenon has been attributed to the myeloid rather than the lymphoid compartment[14], also known as macrophage activation syndrome (MAS). In contrast to lymphocytes, we saw that DCs responded normally to infected cells and were activated in their vicinity. DCs also exhibited a higher tendency to be infected, and myeloid cells were the primary infected cell type among the immune compartments. This observation lends support to the hypothesis that myeloid cells home to the lung[26, 27], get infected, and subsequently serve as a conduit for viral spread to other organs[28, 29]. Although viral replication within the cell might be limited, viral infected myeloid cells have the capacity to induce a great variety of pro-inflammatory cytokines and chemokines, which contribute to MAS[30]. Therefore, as a result of exposure to viral antigens, DCs

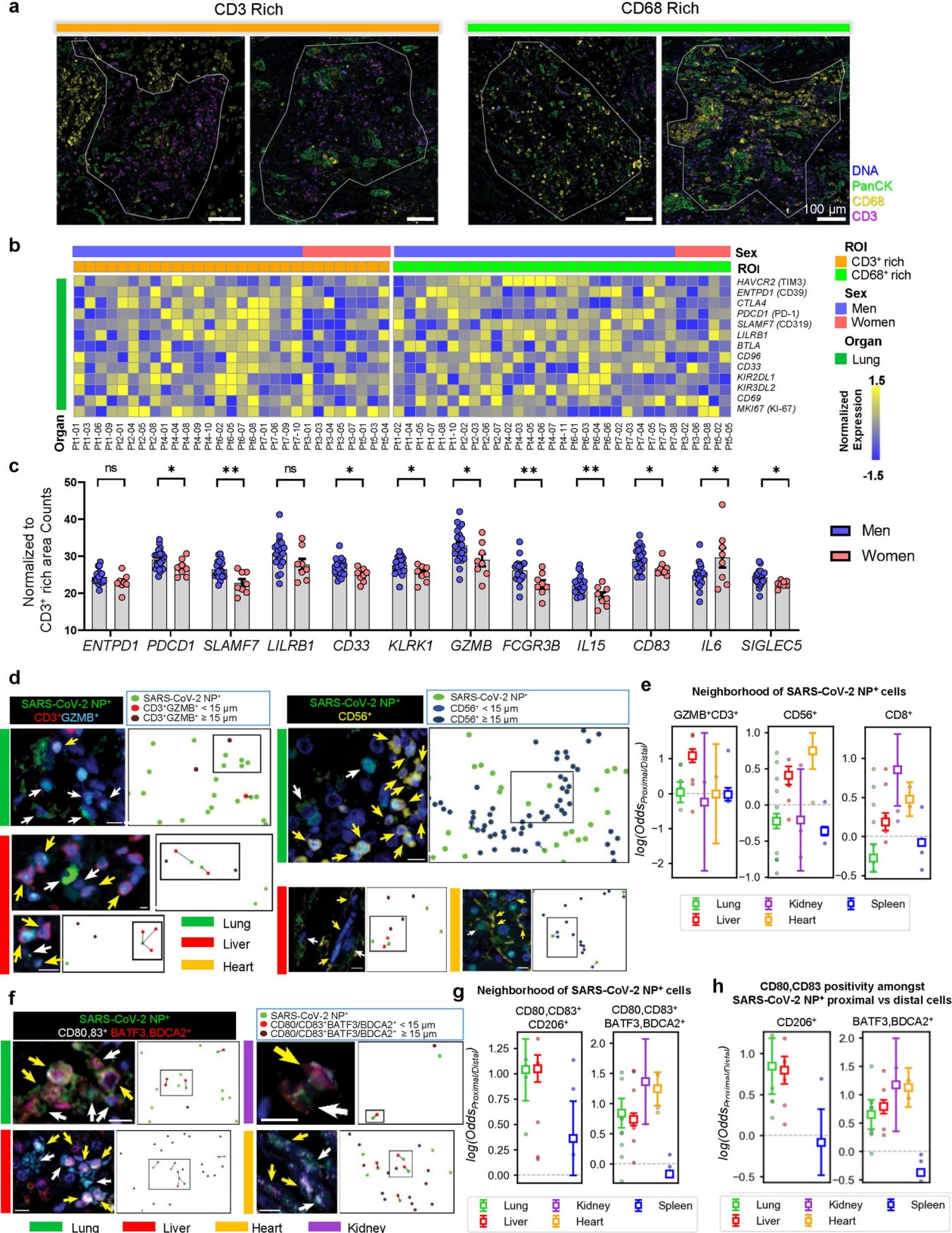

and macrophages proximal to viral antigens have a higher tendency to be activated than distal DCs and macrophages. Ultimately, MAS corresponds to our model whereby such differential contributions originate from the greater proximity of myeloid cells than lymphocytes to infected cells.

Finally, myeloid cells such as monocytes, macrophages and DCs have been reported to express ACE2[30–32]. As such, myeloid cells are indeed susceptible to direct SARS-CoV-2 infection. In particular, multiple IHC studies have clearly identified macrophages in the alveolar septum and lung cavity[33–38]. Positive staining for SARS-

**Fig. 3 Immunosuppression operates on NK cells and T cells but not on myeloid cells. a** Representative CD3[+] ($n = 29$) and CD68[+] ($n = 31$) lung regions profiled by DSP. **b** Normalized gene expression of immunosuppression-related genes in CD3[+] rich and CD68[+] rich ROIs ($n = 60$ ROIs from $n = 7$ decedents, men = 2, women = 5) of lung. Z-scale from −1.5 to 1.5. **c** Comparison of selected genes expression in CD3[+] rich regions between sexes from lung tissues. Each point represents one ROI. Colored dots represent different samples. Blue ($n = 21$), red ($n = 8$). Data are shown as mean ± s.e.m. $p$ (PDCD1) = 0.049; $p$ (SLAMF7) = 0.001; $p$ (CD33) = 0.03; $p$ (KLRK1) = 0.02; $p$ (GZMB) = 0.049; $p$ (FCGR3B) = 0.009; $p$ (IL15) = 0.007; $p$ (CD83) = 0.04; $p$ (IL6) = 0.02; $p$ (SIGLEC5) = 0.04; Unpaired two-tailed Student's $t$ tests, $*p < 0.05$, $**p < 0.01$, $***p < 0.001$, $****p < 0.0001$, ns: not significant. **d, f** mIHC staining of GZMB[+] CD3[+] T cells (**d**, left), CD56[+] NK cells (**d**, right), or CD80[+]/CD83[+] BATF3[+]/BDCA2[+] DCs (**d**) Number of cells depicted/total observed: SARS-CoV-2 NP[+] (Lung 5/1923, Liver 3/702, Heart 1/374); GZMB[+]CD3[+] (Lung 1/723, Liver 7/695); CD56[+] (Lung 13/8783, Liver 3/3802, Heart 4/1077)· (**f**) in the neighborhood of SARS-CoV2-NP+ cells in various organs and corresponding spatial annotations. Number of cells depicted/total observed: SARS-CoV-2 NP[+] (Lung 4/2701, Liver 3/1034, Kidney 1/103, Heart 4/638); CD80,CD83[+] BATF3,BDCA2[+] (Lung 3/796, Liver 6/2792, Kidney 1/4, Heart 2/374). Scalebars: 10 μm. **e, g** Odds ratios of GZMB[+] CD3[+] T cells (**e**, left) (Lung: 0.027 ± 0.295; Liver: 1.085 ± 0.195; Kidney: −0.243 ± 1.977; Heart: −0.014 ±1.430; Spleen: −0.026 ± 0.188), CD56[+] NK cells (**e**, middle) (Lung: −0.227 ± 0.101; Liver: 0.405 ± 0.125; Kidney: −0.209 ± 0.704; Heart: 0.745 ± 0.248; Spleen: −0.364 ± 0.064), CD8[+] cells (**e**, right) (Lung: −0.276 ± 0.177; Liver: 0.090 ± 0.075; Kidney: 0.850 ± 0.459; Heart: 0.477 ± 0.215; Spleen: −0.077 ± 0.039), activated macrophages (**g**, left) (Lung: 1.037 ± 0.303; Liver: 1.048 ± 0.134; Spleen: 0.361 ± 0.366), or activated DCs (**g**, right) (Lung: 0.844 ± 0.243; Liver: 0.736 ± 0.104; Kidney: 1.364 ± 0.706; Heart: 1.242 ± 0.273; Spleen: −0.162 ± 0.039) in NP-proximal compared to NP-distal regions in various organs. **h** Odds ratios of CD80/CD83 positivity in NP-proximal CD206[+] cells compared to NP-distal CD206[+] cells (Lung: 0.844 ± 0.339; Liver: 0.796 ± 0.165; Spleen: −0.083 ± 0.403) and in NP-proximal BATF3/BDCA2[+] cells compared to NP-distal BATF3/BDCA2[+] cells (Lung: 0.652 ± 0.257; Liver: 0.786 ± 0.119; Kidney: 1.170 ± 0.820; Heart: 1.126 ± 0.341; Spleen: −0.376 ± 0.047). Odds ratios of CD80/83 positivity defined as $(N_{(p)CD80/CD83+}/n_{(d)\ CD80/CD83+})/(n_{(p)\ CD80/CD83−}/n_{(d)\ CD80/CD83−})$, where (p) and (d) refer to CD206+ or BATF3/BDCA2[+] cells proximal or distal from viral NP. **e, g, h** Circles indicate sample odds ratios, squares with error bars indicate aggregated odds ratios and confidence intervals. Proximity is as defined by a distance cut-off of 15 μm from viral NP. Data are shown as Odds Ratio ± 95% CI. Source data are provided as a Source Data file.

CoV-2 viral antigen on ACE2[+] macrophages but not T cells were reported[29], as well as flow cytometry analysis indicating SARS-CoV-2 spike protein interaction with ACE2[+] macrophages, but not T cells[32]. It has been reported that monocytes undergo morphological and phenotypic alterations during COVID-19 infection[39, 40], however the effects on DCs are less studied[31]. In this study, we observed that BATF3[+]BDCA2[+] DCs were less abundant in women than in men. We also described an increased likelihood of ACE2[+] cells to be infected over ACE2[−] cells. While ACE2 has been well-characterized as the receptor for the viral spike protein[41], to our knowledge, ours is the first study to relate ACE2 expression to infection rate. Reports have suggested that the decreased COVID-19 severity in women might be related to the ACE2 locus being on the X chromosome[42], however the interplay between DC abundance, ACE2 expression and the risk of cell infection and consequent disease severity remain to be further explored.

Despite our high-dimensional analysis, one limitation of our study is the sample size of 22 decedents, which given the rising global death toll, may now seem like a small number. Our samples originated from the main cohort of the limited number of decedents autopsied during the first outbreak of this pandemic in Wuhan, China. In the initial phase, autopsies were not favored due to onerous safety requirements and insufficient political support[3]. Hence, as a result of the sample size limitation, some statistical tests are under-powered, and we therefore sought to mitigate this shortcoming by profiling the samples through multiple modalities.

While the vast majority of studies that profiled COVID-19 immune responses have done so with peripheral blood samples[2] and low throughput techniques (Supplementary Table 1) which only gather information from a single tissue slice, these are inadequate in obtaining a complete picture of the heterogenous immune landscape and lack the ability to delineate the underlying pathophysiological mechanisms of COVID-19. By comparison, our sample size is comparable to the current state of published COVID-19 autopsy reports, where only 2 multi-organ autopsies using IHC have included more than 10 patients (Supplementary Table 1). Finally, to our knowledge ours is the first study of the in situ immune microenvironment in different tissues during COVID-19 disease.

By combining transcriptome analyses with DSP and mIHC, we have achieved high-dimensional immunopathological profiling of COVID-19. From the resulting data, we propose that severe COVID-19 incites robust immune responses from myeloid cells while selectively suppressing lymphocytes, recapitulating in the tissue context COVID-19-associated observations such as MAS and lymphopenia. The distinct patterns of severe immunosuppression in multiple organs suggest that TIM-3 and PD-1-mediated immunosuppression is a hallmark of severe COVID-19, and that the extent of immunosuppression correlates with male sex and advanced age. At the protein level, PD-1[+] cells can be found in close proximity to TIM-3[+] cells. Activated myeloid cells, but not lymphocytes, were observed in proximity to SARS-CoV-2 viral antigen, thus providing direct pathological evidence for virus-induced myeloid cell activation that likely drives disease progression. Going forward, we propose that the two immune compartments should be considered separately when investigating disease progression.

## Methods

**Study cohort**. The autopsy samples from 22 patients with severe COVID-19 who had undergone autopsy were obtained from the Huoshenshan Hospital, Wuhan, China. All safety precautions were in line with recently published guidelines[43]. All patients provided informed consent and voluntary donation certificates. The age of the patients ranged from 51 to 88 years, with a median age of 66 years. The three lung specimens of non-COVID-19 patients used in the bulk RNA-sequencing analysis were obtained from patients suffering from lung squamous cell carcinoma, pneumonia, and pulmonary bronchiectasis, respectively. The age of these patients ranged from 51 to 73 years, with a median age of 64.0 years. The clinical characteristics of all patients are provided in Supplementary Table 2. Bulk RNA-sequencing, Nanostring Digital Spatial Profiling and multiplex-immunohistochemistry were performed on lung, liver, heart, kidney, and salivary gland specimens from these patients. The specific uses of each autopsy sample used in each experiment are provided in Supplementary Table 4. All samples were anonymously coded in accordance with the Helsinki Declaration.

**Study approval**. This study was approved by the Ethics Committee of the First Affiliated Hospital of Army Medical University (KY2020027) and the First Affiliated Hospital of University of Science and Technology of China (2021KY193). Written informed consent was obtained from all patients. Family members' consent for donation was obtained from all medicine autopsies on patients with COVID-19 infection. Brain and spinal cord samples were not taken due to the need to use a bone saw and the risk of excess aerosolization.

**Bulk RNA sequencing**. The raw RNA sequencing data were subjected to vigorous quality control (QC) procedures before downstream analysis. Raw fastq files were processed with Hisat2 software (v7.5.0). FastQC (v0.11.9) were also used to evaluate the sequencing quality. The clean data were compared using Hisat2[44]. The sequence reads were aligned to the human genome hg19 (GRCh38), and the aligned reads were

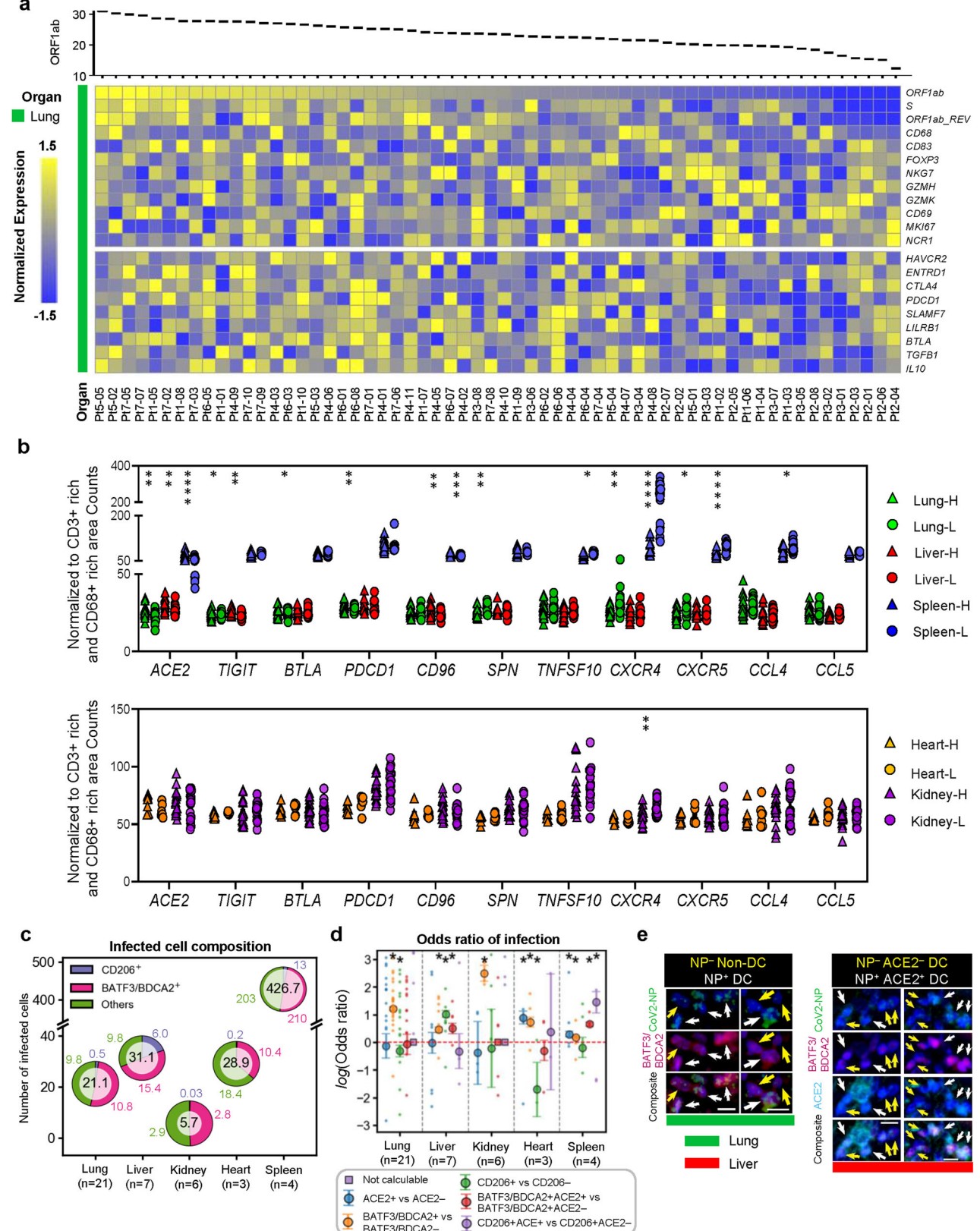

counted within exons using FeatureCounts (v2.0.1) with default parameters to generate the expression matrices of the raw read counts. For the raw data, the rowSums () function in the base package was used that filters out data with read counts < 2. The DESeq2 (v1.26.0) and edgeR (v3.28.1) package in Rstudio (https://rstudio.com/) was used to identify differentially expressed genes. All analyses were performed in R (https://www.r-project.org/;v3.6.1). Heat maps were drawn using Pheatmap(v1.0.12) and the GSEA and GO analysis of the enriched genes was performed using topGO

(version2.38.1), clusterProfiler (version3.14.3), pathview (version1.26.0), AnnotationHub (version2.18.0), org.Hs.eg.db (version3.10.0), tidyverse (version1.3.0), GSEA-Base (version1.48.0), Rgraphviz (version2.30.0), showtext (version0.9), readr (version1.3.1), stats (version3.6.1), enrichplot (version1.6.1). Ggplot2 (version3.3.2), ggrepel (version0.8.2), RcolorBrewer (version1.12) were used drawing and color matching. TrimGalore (https://github.com/FelixKrueger/TrimGalore/issues/25) was used for data QC to remove joints in the pre-processing.

**Fig. 4 Landscape of viral infection and immunosuppression across multiple organs affected by COVID-19. a** Expression of immune-related genes in 60 ROIs from 7 decedents (men = 2, women = 5) in lung tissues, which arranged according to the expression of SARS-CoV-2 (*ORF1ab*). **b** Comparison of the expression of various genes between virus-high and virus-low ROIs in multiple organs. Lung-H (ROIS of high virus express in lung), Lung-L (ROIS of low virus express in lung) Similar to other organs. Each point represents one ROI. Colored dots represent different samples, lung (green, Lung-H = 30, Lung-L = 30), heart (yellow, Heart-H = 8, Heart-L = 7), kidney (purple, Kidney-H = 20, Kidney-L = 20), liver (red, Liver-H = 20, Liver-L = 20) and spleen (blue, Spleen-H = 20, Spleen-L = 20). Data are shown as mean ± s.e.m. *p*-lung (*ACE2*) = 0.003; *p*-lung (*TIGIT*) = 0.03; *p*-lung (*BTLA*) = 0.03; *p*-lung (*PDCD1*) = 0.002; *p*-lung (*SPN*) = 0.003; *p*-lung (*CXCR4*) = 0.003; *p*-lung (*CXCR5*) = 0.02; *p*-liver (*ACE2*) = 0.003; *p*-liver (*TIGIT*) = 0.005; *p*-liver (*CD96*) = 0.005; *p*-spleen (*ACE2*) = 0.00003; *p*-spleen (*CD96*) = 0.0006; *p*-spleen (*TNFSF10*) = 0.01; *p*-spleen (*CXCR4*) < 0.000001; *p*-spleen (*CXCR5*) = 0.000001; *p*-spleen (*CCL4*) = 0.02; *p*-kidney (*CXCR4*) = 0.006; *p < 0.05, **p < 0.01, ***p < 0.001, ****p < 0.0001, ns: not significant: Unpaired two-tailed Student's *t* tests. **c** Average number of NP$^+$ cells per ROI in each organ and corresponding myeloid composition. **d** Odds ratios of infection of various cell types in each organ, defined as (N$_{(x)}$ NP+/N$_{(y)}$NP+)/(N$_{(x)}$NP−/N$_{(y)}$NP−), where (x) and (y) refer to two cell types compared to each other. When the odds ratio is close to 1, both cell types are as likely to be NP$^+$. Dots indicate sample odds ratios, squares with error bars indicate aggregated odds ratios and confidence intervals. *Indicates that confidence interval, *p*-lung (BATF3,BDCA2$^+$ vs BATF3,BDCA2$^-$) < 0.001, *p*-lung (CD206$^+$ vs CD206$^-$) = 0.004; *p*-liver (BATF3,BDCA2$^+$ vs BATF3,BDCA2$^-$) < 0.001, *p*−liver (CD206$^+$ vs CD206$^-$) < 0.001, *p*-liver (BATF3,BDCA2$^+$ACE2$^+$ vs BATF3,BDCA2$^+$) = 1.5E-07; *p*-kidney (BATF3,BDCA2$^+$ vs BATF3,BDCA2$^-$) < 0.001; *p*-heart (ACE2$^+$ vs ACE2$^-$) = 5.97E-14, *p*-heart (BATF3,BDCA2$^+$ vs BATF3,BDCA2$^-$) = 1.11E-16, *p*-heart (CD206$^+$ vs CD206$^-$) = 2.09E−04; *p*-spleen (ACE2$^+$ vs ACE2$^-$) = 4.48E-10, *p*-spleen (BATF3,BDCA2$^+$ vs BATF3,BDCA2$^-$) < 0.001, *p*-spleen (BATF3,BDCA2$^+$ACE2$^+$ vs BATF3·BDCA2$^+$) < 0.001, *p*-spleen (CD206$^+$ACE2$^+$ vs CD206$^+$) = 7.16E-14. (refer to Methods for statistical test performed). **e** mIHC staining of DCs *p*ositive (white arrows) or non$^-$DCs negative (yellow arrows) for NP (left), and ACE2$^+$ DCs positive (white arrows) or ACE2$^-$ DCs negative (yellow arrows) for NP. Number of SARS-CoV-2 NP$^+$ DCs depicted/total observed in lung: 5/1288. Number of SARS-CoV-2 NP$^+$ ACE2$^+$ DCs depicted/total observed in liver: 4/82. Scalebars: 10 μm. Source data are provided as a Source Data file.

**Digital spatial profiling (DSP).** NanoString spatial profiling was used for DSP analysis of three protein-level and 1800 transcriptomic-level immune-markers simultaneously on FFPE tissue slides[45, 46]. Following deparaffinization and antigen retrieval procedures, sections were simultaneously incubated overnight with fluorescent-labeled antibodies against CD3 (Origene), CD68 (Santa Cruz) and Pan-Cytokeratin (Novus Biologicals), smooth muscle actin (Invitrogen), CD20 (Novus Biologicals), or cytokeratin 8/18 (Novus Biologicals) to visualize morphological features in the regions of interest (ROI). After staining, the tissues were scanned using a GeoMx DSP instrument to generate digital fluorescent images and to select for individual ROIs inside a geometric section. To carry out high-resolution multiplex profiling, each ROI was assigned to CD3$^+$ T-cell-rich or CD68$^+$ macrophage-rich regions. UV light was directed through a programmable digital micro mirror device (DMD) or dual DMD (DDMD) to accurately illumine the ROI and cleave PC oligos from the selected region, which were collected by microcapillary tube inspiration and dispensed into a 96-well plate. Inside the microplate, the single-molecule counting nCounter System enabled the digital counting of released oligos[47, 48].

**Digital spatial profiling (DSP) analysis.** Q3 Normalization – Individual counts were normalized against the 75th percentile of the signal from their own ROI[49]. We scale our AOIs so that they all have the same value for their Quartile 3 value. In particular, we (i) first, divide all the genes per AOI by their respective Q3 count, and (ii) second, multiply all the genes in all AOIs by a constant, defined as the geometric mean of Q3 counts for all AOIs. This approach generally performs well for CTA data and is often the preferred normalization. Normalize. Quantiles is used to correct the different batch.

**Immunohistochemistry (IHC).** IHC was performed on FFPE tissue samples as previously described[50–52]. Lung, liver, heart, kidney, and spleen (4-μm thick) were labeled with antibodies targeting the SARS-CoV-2 NP (Supplementary Table 3). Appropriate controls were included. Images were captured using an IntelliSite Ultra-Fast Scanner (Philips, Eindhoven).

**Multiplex immunohistochemistry (mIHC).** mIHC was performed using an Opal Multiplex fIHC kit (Akoya Biosciences, California). FFPE tissue sections were cut onto Bond Plus slides (Leica Biosystems, Richmond) and heated at 60 °C for 20 min. The tissue slides were then subjected to deparaffinization, rehydration and heat-induced epitope retrieval using a Leica Bond Max autostainer (Leica Biosystems, Melbourne) before endogenous peroxidase blocking (Leica Biosystems, Newcastle). Next, the slides were incubated with primary antibodies followed by incubation with polymeric HRP-conjugated secondary antibodies (Leica Biosystems, Newcastle) (Supplementary Table 3). Then, the samples were incubated with Opal fluorophore-conjugated tyramide signal amplification (TSA) (Akoya Biosciences, California) at a 1:100 dilution. The slides were rinsed with wash buffer (BOND Wash Solution 10X Concentrate) after each step. Following TSA deposition, the slides were again subjected to heat-induced epitope retrieval to strip the tissue-bound primary/secondary antibody complexes prior to further labeling. These steps were repeated until the samples were labeled with all six markers and spectral DAPI (Akoya Biosciences, California) at a 1:10 dilution. Finally, the slides were mounted in ProLong Diamond Anti-fade Mountant (Molecular Probes, Life Technologies, USA) and developed in the dark at room temperature for 24 h.

**mIHC analysis.** Multiple random ROIs were scored for each patient sample. Analyses were performed using Python(v3.8.2) on the cell-segmented output from the HALO software and all mIHC analysis plots were created using the matplotlib

(v3.2.1) package. Statistical tests were performed using the scipy (v1.3.2), numpy (v1.18.5), pandas (v1.0.5) and statsmodels (v0.11.1) packages. Populations proximal or distal from "central cell" markers (TIM-3$^+$ or SARS-CoV-2 NP$^+$ in this paper) were determined by calculating a distance matrix between the centroids of each central cell to every other cell and identifying those cells <15 μm from a central cell (except the central cell itself) as proximal and all other cells as distal. ROIs from the same sample were combined by summation of proximal and distal cell counts. ROIs without central cells were disregarded. Odds ratios were aggregated to derive aggregated odds ratios and confidence intervals using Cochran-Mantel-Haenszel and Breslow-Day procedures as implemented by the Stratified Table class of the statsmodels module using default values. Comparison of cell percentages between sexes in Fig. 2d and Supplementary Fig. 2e were performed by averaging of cell percentages across all ROIs of each patient, followed by taking the average of those percentages in men and dividing by the average of those percentages in women, and transforming the resulting by log$_2$. Odds ratios defined as (N$_{(x)NP+}$/N$_{(y)NP+}$)/(N$_{(x)NP}$$_-$/N$_{(y)}$NP−), where (x) and (y) refer to two cell types compared to each other. When the odds ratio is close to 1, both cell types are as likely to be NP$^+$ in Fig. 4d

**Reporting summary.** Further information on research design is available in the Nature Research Reporting Summary linked to this article.

## Data availability
The RNA-Seq and GeoMx data generated in this study have been deposited in the National Genomics Data Center database under the accession numbers HRA000974, OMIX488 and Gene Expression Omnibus under accession numbers GSE182917, GSE182920. The images can be queried in the database BioStudies under accession number S-BIAD170. Source data are provided with this paper.

## Code availability
Code for the analysis of RNA-seq data can be accessed via: https://github.com/peiqi-sudo/Postmortem-high-dimensional-immune-profiling-of-severe-COVID-19-patients-reveals-distinct-patterns-o.

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

## Acknowledgements

This work was supported by the National Natural Science Foundation of China (#92169118, #8202290021), National Key Research and Development Program of China (#2021YFC2300600), Anhui Provincial Natural Science Foundation (#2008085J35, 1908085MH281), and the Fundamental Research Funds for the Central Universities (WK9110000055).

## Author contributions

C.S. conceived and directed the study. H.B.W., Y.R., S.Q.X., H.L., W.W., Z.B.L., D.Y.Z., J.C., X.D.Z., D.P.J., X.C.F., L.Z., H.Z., Z.H.L., R.C., W.Q.L., C.F.W., S.Y.Z., J.W.Q., B.N. and Z.W. collect autopsy specimens. P.Q.H. and H.B.W. performed and interpreted the IHC data. Y.R. and Z.B. provide IHC technical support. P.Q.H. and H.B.W. performed biostatistical analysis. P.Q.H. analyzed Nanostring DSP profiling. C.S. drafted the manuscript and final approval of all authors.

## Competing interests

The authors declare no competing interests.
