## [Peer Review File · Nature Communications]

REVIEWER COMMENTS

Reviewer #1 (Viral immunity, immune phenotyping) (Remarks to the Author):

He et al. provide a detailed analysis of tissues from 22 autopsies of patients who died of COVID-19 in Wuhan, China. They performed bulk RNAseq, digital spatial profiling of transcripts, and multiplex immunohistochemistry to identify cell types. Tissues included lung, liver, heart, kidney, and salivary gland. Among their major findings are sex-specific differences in immunosuppression levels (e.g., TIM-3 and PD-1 levels); proximity of activated myeloid (not lymphoid) cells to viral antigens; and correlations of viral load with ACE2 and immunosuppressive markers. These findings both corroborate and advance previous knowledge about the pathology of SARS-CoV-2 infection. There have been relatively few studies of autopsy tissues in COVID-19, and none with this level of comprehensive phenotyping, making this an important study and data resource.

The main issue I see with the study as presented is the limited amount of organ-specific data. With the exception of Figure 4, the authors present their findings with little mention of the organs from which those findings derived. For example, Figure 2 does not identify the tissue type analyzed, or whether findings shown are consistent across different organs. More detailed description of concordance or discordance across organs for these data would be helpful to understand the generalizability of sex differences in tissue findings. Similarly, please clarify if all data in Figure 1 is from lung tissue, and if so, might it be possible to show whether other organs show similar or discordant findings?

Reviewer #2 (Systems immunology, single cell analyses) (Remarks to the Author):

The authors applied RNA-seq, IHC, and DSP to COVID-19 autopsy samples and presented an insightful story about the immune status of patients. While the data and the story are interesting, the presentation and writing of this manuscript are poor, prohibiting readers from understanding the logic underlying the authors' claims.

Major concerns:

1. Figures legends are either not properly matched to figures or too simplified. Readers cannot get the details of each plot. For example, Figure 1A is a PCA plot but the legend is "A, Differential gene expression in lung tissue of COVID-19 (n=11) compared to non-COVID-19 (n=3) decedents". Figure 1B is for differential expression analysis but the legend is "B, C, Principal component analysis (PCA) (B) and gene set enrichment analysis (GSEA) (C) of decedents in (A)". It is hard to know how those genes were ranked in Figure 1C. In Figure 1E, what unit is used? In Figure 2D, what is "Relevance abundance"? In Figure 3D and 4F, what are the "normalized data"? In Figure 4C, how is "risk ratio" defined?
2. Since all the samples were autopsies, I cannot understand what the sex differences the authors claimed meant. The authors should clearly clarify the clinical meanings of such comparisons.
3. In Figure 4B, the percentage of SARS-CoV-2+ cells in the liver was much higher than those of other organs including the lung. The authors should elaborate on this point further, including the types of SARS-CoV-2+ cells in these organs.
4. In Figure 4D, markers for dendritic cells should be provided.
5. In Figure 2A, 3A, and 4E, markers for SARS-CoV-2 should be provided.

Overall, while this manuscript reported multiple interesting findings, the data presented in figures and the corresponding annotation and explanation did not support sufficiently these claims. The clinical implications are also unclear. Major revisions are needed to further substantiate these interesting findings.

Reviewer #1 (Viral immunity, immune phenotyping) (Remarks to the Author):

He et al. provide a detailed analysis of tissues from 22 autopsies of patients who died of COVID-19 in Wuhan, China. They performed bulk RNAseq, digital spatial profiling of transcripts, and multiplex immunohistochemistry to identify cell types. Tissues included lung, liver, heart, kidney, and salivary gland. Among their major findings are sex-specific differences in immunosuppression levels (e.g., TIM-3 and PD-1 levels); proximity of activated myeloid (not lymphoid) cells to viral antigens; and correlations of viral load with ACE2 and immunosuppressive markers. These findings both corroborate and advance previous knowledge about the pathology of SARS-CoV-2 infection. There have been relatively few studies of autopsy tissues in COVID-19, and none with this level of comprehensive phenotyping, making this an important study and data resource. The main issue I see with the study as presented is the limited amount of organ-specific data.

Reply: Thank you for your kind advice and this point is well taken in our revised edition. Indeed, more organ-specific data will help readers to better understand and follow. According to your suggestion, we have done our best to increase the sample size in both bulk-RNA sequencing data and digital spatial profiling.

For bulk-RNA sequencing, 10 new cases (4 cases of liver [Pt2, Pt3, Pt5, Pt7], 2 cases of heart [Pt5, Pt6], 2 cases of spleen [Pt5, Pt6], and 2 cases of kidney [Pt3, Pt6]) were added into the characteristics of immune status of male and female patients with severe COVID-19 infection. By adding more samples, we have verified the findings of differences between men and women, and we found that the immunosuppression signature was more pronounced in men compared to women (Fig. 1d-f).

For digital spatial profiling, 9 new cases (2 cases of liver [PT1, PT4], 1 case of heart [Pt4], 2 cases of kidney [Pt2, Pt16], and 4 cases of spleen [Pt1, Pt2, Pt6, Pt16]) were added to further analysed the spatial immune profiling of different organs of patients with COVID-19 (Fig. 2-4). Furthermore, mIHC analyses were also performed on the same additional organs in new add 12 cases (1 cases of liver, 6 cases of kidney, 1 cases of heart, 4 cases of spleen), and analyses data has been included to Fig. 2f, 3d-g, 4b-d and Supplementary. Fig. 2c, 5c-d.

Altogether, we have added 31 cases including liver, spleen, heart and kidney data to our revised manuscript to better answer your question. Thank you very much for calling this to our attention to let us verify the conclusion of our manuscript by increasing the number of samples. We have put the information table of added samples and the revised figures below for your convenience. We have also revised our Results and Discussion sections accordingly (page 3, line 19; page 4, line 11; page 5, line 8; page 6, line19; page 7, line18).

Table: Overview of the experimental use for newly added tissue samples.

	Liver	Heart	Spleen	Kidney	Total
DSP	2	1	4	2	9
Bulk	4	2	2	2	10
mIHC	1	1	4	6	12
Total	7	4	10	10	31

Figure 1: **(d)** Principal component analysis of bulk transcriptomes of lung (green, women = 7, men = 4), heart (yellow, women=1, men=1), kidney (purple, women=1, men=1), liver (red, women=2, men=2) and spleen (blue, women=1, men=1). **(e)** Comparison chemokine expression levels between men and women in lung, heart, kidney, liver and spleen. Colored according to the sample based on RNA-seq data. **(f)** Gene Ontology analysis of pathways enriched of multiple organs in women upregulate compared to men. Colored according to the sample based on RNA-seq data.

Figure 2: **(f)** Odds ratio of TIM-3⁺ cell-proximal compared to TIM-3⁺ cell-distal PD-1⁺ cells in

various organs. Odds ratio defined as $(N(p)PD1^+/N(d)PD1^+)/((N(p)PD1^-/N(d)PD1^-))$, where (p) and (d) refer to cells proximal or distal from TIM-3⁺ cells. Proximity is as defined by a distance cut-off of 15µm. Dots indicate sample odds ratios, squares with error bars indicate aggregated odds ratios and confidence intervals.

d

e

Figure 3

f

g

Figure 3: **(d, f)** mIHC staining of GZMB⁺ CD3⁺ T cells (d, left), CD56⁺ NK cells (d, right), or CD80⁺/CD83⁺ BATF3⁺/BDCA2⁺ DCs (f) in the neighbourhood of SARS-CoV2-NP⁺ cells in various organs and corresponding spatial annotations. Scalebars: 10 µm. **(e, g)** Odds ratios of GZMB⁺ CD3⁺ T cells (e, left), CD56⁺ NK cells (e, middle), CD8⁺ cells (e, right), or activated myeloid cells (g), in NP-proximal compared to NP-distal regions in various organs.

b

Figure 4

Figure 4: **(b)** Comparison of the expression of various genes between virus-high and virus-low ROIs in multiple organs. Lung-H (ROIs of high virus express in lung), Lung-L (ROIs of low virus express in lung) Similar to other organs. * $p < 0.05$, ** $p < 0.01$, *** $p < 0.001$, **** $p < 0.0001$, ns: not significant): two-tailed Student's t-tests. **(c)** Average number of NP+ cells per ROI in each organ and corresponding myeloid composition. **(d)** Odds ratios of infection of various cell types in each organ, defined as $(N(x)NP^+ / N(y)NP^+) / (N(x)NP^- / N(y)NP^-)$, where (x) and (y) refer to two cell types compared to each other. When the odds ratio is close to 1, both cell types are as likely to be NP+. Dots indicate sample odds ratios, squares with error bars indicate aggregated odds ratios and confidence intervals

Supplementary Figure 2

Supplementary. Figure 2: **(c)** Comparison of percentage of TIM-3⁺ (left) or PD-1⁺TIM-3⁺ (right) cells in lung tissue between men (n=11) and women (n=10). p-values: two-tailed Student's t-test.

With the exception of Figure 4, the authors present their findings with little mention of the organs from which those findings derived.

Reply: Thank you very much for your suggestion, and we apologize for our negligence. We have added the source organs of our finding in all figures according to your suggestion. To make it easier to distinguish, we have added different colours to mark different organ sources (green represents the lungs, orange represents the heart, purple represents the kidneys, blue represents the spleen, and red represents the liver). We have also examined our manuscript carefully, and revised some errors to make sure this mistake will not happen throughout the manuscript.

For example, Figure 2 does not identify the tissue type analyzed, or whether findings shown are consistent across different organs. More detailed description of concordance or discordance across organs for these data would be helpful to understand the generalizability of sex differences in tissue findings.

Reply: Thank you for your kind advice. We have added corresponding marks to identify the tissue type used to all figures including Figure 2 according to your advice. We have revised the manuscript accordingly (page 21, line 17) to ensure that the corresponding organ data is properly labelled to allow clearer visualisation.

To better understand the generalizability of sex differences in tissue findings, we have added comparisons of male and female differences in liver, spleen, kidney and heart to Fig. 2b-c and Supplementary. Fig. 2a-e. Similar to the findings in lungs from Figure 2, there were obvious differences between men and women in the spleen, kidneys, and heart (Supplementary. Fig. 2a); immune responses shown in tissue regions of interest obtained from women were more active than men (Supplementary. Fig. 2b). Interestingly, we were not able to detect such differences in the liver, suggesting that the immune microenvironment of the liver is different from other organs (Supplementary. Fig. 2). We have revised our Results sections accordingly (page 4, line 17; page 4, line 19; page 4, line 22).

Figure 2

Figure 2: **(b)** Heatmap show normalized gene expression levels across 60 ROIs from 7 COVID-19 decedents (men = 2, women = 5) of lung. Z -scale from -1.5 to 1.5. **(c)** Comparison of expression of selected genes between sexes of lung tissues with COVID-19. Unpaired two-tailed Student's t-tests. * $p < 0.05$, ** $p < 0.01$, *** $p < 0.001$, **** $p < 0.0001$, ns: not significant.

Supplementary Figure 2

Supplementary. Figure 2: **(a)** Heatmap showing the differential gene expression between men (n=4 and women (n=2) from liver (red, 40 ROIs from 4 decedents), heart (orange, 15 ROIs from 2 decedents), spleen (blue, 40 ROIs from 4 decedents) and kidney (purple, 40 ROIs from 4 decedents). **(b)** Comparison of selected genes expression between sexes from liver (red, 40 ROIs from 4 decedents), heart (orange, 15 ROIs from 2 decedents), spleen (blue, 40 ROIs from 4 decedents) and kidney (purple, 40 ROIs from 4 decedents). Unpaired two-tailed Student's t-tests, * $p < 0.05$, ** $p < 0.01$, *** $p < 0.001$, **** $p < 0.0001$. **(c)** Comparison of percentage of TIM-3⁺ (left) or PD-1⁺TIM-

3⁺ (right) cells in lung tissue between men (n=11) and women (n=10). p-values: two-tailed Student's t-test. **(d)** Differences in percentages of TIM-3⁺ cells (left) or PD-1⁺TIM-3⁺ cells (right) between men and women in various organs. **(e)** Correlation between age and percentage of TIM-3⁺ cells or TIM-3⁺PD-1⁺ cells in lung tissue of men (top) and women (bottom). p-values: two-tailed Wald test.

Similarly, please clarify if all data in Figure 1 is from lung tissue, and if so, might it be possible to show whether other organs show similar or discordant findings?

Reply: Thank you for your kind advice and this point is well taken in our revised edition. We have added the source of tissues in all figures, legends and results according to your suggestion. For bulk-RNA sequencing, our last version of the manuscript only had sequencing data from lung tissues. To confirm whether other organs show similar or discordant findings from lung tissue, we have added a total of 10 new cases of bulk-RNA sequencing data, including 4 cases of liver [Pt2, Pt3, Pt5, Pt7], 2 cases of heart [Pt5, Pt6], 2 cases of spleen [Pt5, Pt6], and 2 cases of kidney [Pt3, Pt6] into Fig. 1d, e, f. Analysis of bulk-RNA data from a total of 24 patients found that in multiple organs (liver/heart/spleen/kidney), immune-associated gene expressions differed between men and women, with a trend suggesting that women had a more activated immune response than men. These findings suggested that differences between men and women are common in the lungs, liver, kidneys and spleen. The newly added figure has been provided in the answer above. We have also revised our manuscript accordingly (page 3, line19). Thank you very much for calling this to our attention.

Reviewer #2 (Systems immunology, single cell analyses) (Remarks to the Author):

The authors applied RNA-seq, IHC, and DSP to COVID-19 autopsy samples and presented an insightful story about the immune status of patients. While the data and the story are interesting, the presentation and writing of this manuscript are poor, prohibiting readers from understanding the logic underlying the authors' claims.

1. Figure legends are either not properly matched to Figures or too simplified. Readers cannot get the details of each plot. For example, Figure 1A is a PCA plot but the legend is "A, Differential gene expression in lung tissue of COVID-19 (n=11) compared to non-COVID-19 (n=3) decedents". Figure 1B is for differential expression analysis but the legend is "B, C, Principal component analysis (PCA) (B) and gene set enrichment analysis (GSEA) (C) of decedents in (A)". It is hard to know how those genes were ranked in Figure 1C. In Figure 1E, what unit is used? In Figure 3D and 4F, what are the "normalized data"? In Figure 2D, what is "Relevance abundance"? In Figure 4C, how is "risk ratio" defined?

Reply: Thank you for your careful review and pointing out the ambiguities in the Figure legends. According to your advice, we have thoroughly checked through the manuscript and made all necessary amendments to clearly describe the Figures (page 21, line 1). In Fig. 1c Gene set enrichment analysis of hallmark lymphocyte inhibitory genes based on non-COVID-19 to COVID-19. FDR <0.05, |NES| >1 defined as significant. More detail please refer to Source data. In the previous version of Fig. 1e unit refers to after DESeq2 difference analysis, the normalized value is obtained, and then log2 processing is performed to obtain the data for drawing the heatmap. In the revised version the unit is the multiple of log2 fold-change of the differential gene expression in men and women samples. The specific gene expression of different organs is obtained through DESeq2 (Lung/Liver) and edge R (kidney/heart/spleen). We have also added corresponding description in the method section in our revised manuscript (page 12, line 12). In the previous version of Fig. 3d and 4f (The numbers in the revised version are Fig. 3c and Fig. 4b), the "normalized data" refers to normalized specific gene counts from CD3⁺

rich or CD68⁺ rich area. Individual counts were normalized against the 75th percentile of the signal from their own ROI. To avoid ambiguity, we replaced “normalized data” with "Normalized to CD3⁺/CD68⁺ rich area Counts". The updated figure and legend are shown below. Meanwhile, a detailed description was added in the method section in our manuscript (page 13, line 16). In Fig. 2d, “relative abundance” refers to the comparison between samples from men and women of the percentage of cells that are of a particular phenotype. In particular, this comparison is expressed as a log₂ fold-change. In the previous version of Fig. 4c (The numbers in the update version are Fig. 4d), Regarding the “risk ratio”, we have also selected to express likelihoods by odds ratios instead of risk ratios for consistency across all Figures. The definitions for these terms are now included in the methods section and corresponding Figure legends. We believe that our revised manuscript will allow readers to clearly understand the content of the Figures.

Figure 1

Figure1: (e) Comparison chemokine expression levels between men and women in lung, heart, kidney, liver and spleen. Colored according to the sample based on RNA-seq data

Figure 3
Figure 3: **(c)** Comparison of selected genes expression in CD3⁺ rich regions between sexes from lung tissues. Unpaired two-tailed Student's t-tests, * $p < 0.05$, ** $p < 0.01$, *** $p < 0.001$, **** $p < 0.0001$.

Figure 4
Figure 4: **(b)** Comparison of the expression of various genes between virus-high and virus-low ROIs in multiple organs. Lung-H (ROIs of high virus express in lung), Lung-L (ROIs of low virus express in lung) Similar to other organs. * $p < 0.05$, ** $p < 0.01$, *** $p < 0.001$, **** $p < 0.0001$, ns: not significant): two-tailed Student's t-tests.

Figure 2
Figure 2: **(d)** Comparison of percentages of cells in lung tissue expressing various immune markers between men (n=11) and women (n=10). CoV2-NP: SARS-CoV-2 nucleocapsid protein. p-value: two-tailed Student's t-test (left).

Figure 4: **(d)** Odds ratios of infection of various cell types in each organ, defined as $(N(x)NP^+ / N(y)NP^+) / (N(x)NP^- / N(y)NP^-)$, where (x) and (y) refer to two cell types compared to each other. When the odds ratio is close to 1, both cell types are as likely to be NP⁺. Dots indicate sample odds ratios, squares with error bars indicate aggregated odds ratios and confidence intervals (right).

2. Since all the samples were autopsies, I cannot understand what the sex differences the authors claimed meant. The authors should clearly clarify the clinical meanings of such comparisons.

Reply: Thank you for bring this concern up. Indeed, if the difference in the immune environment between men and women has no clinical significance, then there is no comparative significance. As provided in our original manuscript, we have emphasized the clinical significance of the differences between men and women. Specifically, although these are autopsy specimens, we have compared the overall survival time of these patients. As shown in Supplementary. Fig. 1d, the survival time of female patients is significantly longer than that of male patients (page 4, line 9). In addition, study based on 1,099 patients with COVID-19 and 524 patients with SARS has observed similar findings that men with COVID-19 are more at risk for worse outcomes and death, independent of age (Front Public Health.2020 Apr 29;8:152. doi: 10.3389/fpubh.2020.00152. PMID: 32411652). These differences suggested that the immune

microenvironment of male and female patients may differ, and these results were subsequently proved in Fig. 2 and 3. For details, please refer to (page 4, line 11). Thank you for your understanding.

d

Supplementary Figure 1

Supplementary. Figure 1: **(d)** Kaplan-Meier survival curve of men (n=4) and women (n=7) with COVID-19. p-value: log-rank test.

3. In Figure 4B, the percentage of SARS-CoV-2+ cells in the liver was much higher than those of other organs including the lung. The authors should elaborate on this point further, including the types of SARS-CoV-2+ cells in these organs.

Reply: We thank the reviewer for this observation and the opportunity to elaborate on this. We note that the liver exhibits more infected cells compared to the lung, kidney, and heart. This might be partly attributable to infection of a fraction of liver macrophages. While we can infer the immune cell types of infected cells as we performed co-staining of viral NP with immune markers, unfortunately we are unable to identify the other types of infected cells that do not co-express immune markers. We totally agree with you on the importance of figuring out the type of infected cell. We recently included the spleen into the multi-organ analysis and found that the spleen has more than ten times the number of infected cells compared to the other organs examined and similarly, the major infected population were BATF3/BDCA2⁺ DCs. The specific results have shown below. (The original Fig.4b, now is Fig.4c in the revised manuscript). We appreciate your understanding very much.

Figure 4

Figure 4:(c) Average number of NP⁺ cells per ROI in each organ and corresponding myeloid composition.

4. In Figure 4D, markers for dendritic cells should be provided.

Reply: Thanks for your careful review and pointing up this. We have annotated the DC marker (BATF3/BDCA2⁺) outside of the images for clarity in our revised manuscript (The original Fig.4d, now is Fig.4e in the revised manuscript). The updated figure and legend are shown below.

Figure 4

Figure 4: (e) mIHC staining of DCs positive (white arrows) or non-DCs negative (yellow arrows) for NP (left), and ACE2⁺ DCs positive (white arrows) or ACE2⁻ DCs negative (yellow arrows) for NP. Scalebars: 10 μ m.

5. In Figure 2A, 3A, and 4E, markers for SARS-CoV-2 should be provided.

Reply: Thank you very much for your comment. Please be kindly advised that the Figures referenced refer to tissue staining of morphological markers used for guiding ROI selection for DSP and do not include any markers for SARS-CoV-2. The levels of SARS-CoV-2 in each ROI were only subsequently determinable based on RNA levels of SARS-CoV-2 probes. In order to avoid misunderstandings, we have also added corresponding descriptions in our revised version (page 13, line 23). We have also summarized all the immunohistochemical antibodies used in our manuscript in Supplementary Table 3 for your convenience. Thank you again for your kind advice.

Overall, while this manuscript reported multiple interesting findings, the data presented in Figures and the corresponding annotation and explanation did not support sufficiently these claims. The clinical implications are also unclear. Major revisions are needed to further substantiate these interesting findings.

Reply: Thank you for your kind and detailed suggestions. We have read all your comments very carefully and tried our best to revise the manuscript and follow as many suggestions as possible. We appreciate your understanding very much.

REVIEWERS' COMMENTS

Reviewer #1 (Remarks to the Author):

Wu et al. have significantly revised their manuscript in response to the initial reviews. In particular, they have added additional data--more cases of specific postmortem tissues and comparisons across those tissues, including more samples for RNAseq, multiplex IHC, and digital spatial profiling.

Moreover, the authors now call out (using a consistent coloring scheme) the source organs for data in all their figures. This has greatly increased the interpretability of the findings, and better supports the conclusions of organ-specific differences in infection and immune cell organization.

In the case of sex differences, the authors can now show that there are differences between men and women in multiple tissues, though the extent and direction of these differences vary between lung, liver, kidney, and spleen. This is rather broadly described on p 4 of the revised manuscript; in fact the details are more complex, but readers can now see the full extent of the findings in the figures.

The authors have also made efforts to address the concerns of Reviewer 2. These include improved figure legends, tissue comparisons of infected cell frequency, and better figure labeling.

Reviewer #2 (Remarks to the Author):

The authors have addressed all my concerns. The findings are interesting and important for understanding the pathogenesis of COVID-19.